# The clinical academic workforce of the future: a cross-sectional study of factors influencing career decision-making among clinical PhD students at two research-intensive UK universities

Joana Lopes,[1] Veronica Ranieri,[2,3] Trevor Lambert,[4] Chris Pugh,[1] Helen Barratt,[3] Naomi J Fulop,[3] Geraint Rees,[2] Denise Best[1]

► Prepublication history and additional material are available. To view these files please visit the journal online (http://dx.doi.org/10.1136/bmjopen-2017-016823).

[1]Oxford University Clinical Academic Graduate School, John Radcliffe Hospital (Main Hospital), Oxford, UK
[2]Academic Careers Office, School of Life and Medical Sciences, University College London, London, UK
[3]Department of Applied Health Research, University College London, London, UK
[4]Nuffield Department of Population Health, University of Oxford, Oxford, UK

**Correspondence to**
Joana Lopes;
joana.lopes@medsci.ox.ac.uk

## ABSTRACT

**Objectives** To examine clinical doctoral students' demographic and training characteristics, career intentions, career preparedness and what influences them as they plan their future careers.

**Design and setting** Online cross-sectional census surveys at two research-intensive medical schools in England in 2015–2016.

**Participants** All medically qualified PhD students (N=523) enrolled at the University of Oxford and University College London were invited to participate. We report on data from 320 participants (54% male and 44% female), who were representative by gender of the invited population.

**Main outcome measures** Career intentions.

**Results** Respondents were mainly in specialty training, including close to training completion (25%, n=80), and 18% (n=57) had completed training. Half (50%, n=159) intended to pursue a clinical academic career (CAC) and 62% (n=198) were at least moderately likely to seek a clinical lectureship (CL). However, 51% (n=163) had little or no knowledge about CL posts. Those wanting a CAC tended to have the most predoctoral medical research experience ($\chi^2$ (2, N=305)=22.19, p=0.0005). Key reasons cited for not pursuing a CAC were the small number of senior academic appointments available, the difficulty of obtaining research grants and work-life balance.

**Conclusions** Findings suggest that urging predoctoral clinicians to gain varied research experience while ensuring availability of opportunities, and introducing more flexible recruitment criteria for CL appointments, would foster CACs. As CL posts are often only open to those still in training, the many postdoctoral clinicians who have completed training, or nearly done so, do not currently gain the opportunity the post offers to develop as independent researchers. Better opportunities should be accompanied by enhanced career support for clinical doctoral students (eg, to increase knowledge of CLs). Finally, ways to increase the number of senior clinical academic appointments should be explored since their lack seems to significantly influence career decisions.

## Strengths and limitations of this study

► This is the first UK study to describe the career aspirations of medically qualified PhD students and what influences them as they make their postdoctoral career plans.

► Other studies about the retention of postdoctoral clinical academics are predominantly North American and reliant on retrospective data.

► The study identifies factors that are associated with clinical PhD students' decision to apply for clinical lectureships (a pivotal post in clinical academic training in England).

► As it is a cross-sectional study, causality cannot be inferred from the associations we found between career plans and other variables.

► It is not possible to say whether respondents' intentions will be implemented as planned and findings may not generalise beyond research-intensive universities in England.

## INTRODUCTION

Clinical academics (also known internationally as physician scientists) are vital to educating medical students and, as they bring a clinical perspective to research, to discovering and realising new insights into human disease.[1] Establishing a stable pipeline that develops clinical academics[2] continues to be of global concern, for example in the USA and Australia.[3 4] In the UK, anxieties date back at least to the Savill report[1] in 2000 and the subsequent Modernising Medical Careers report.[5] The establishment of the National Institute for Health Research (NIHR) in 2006 and similar developments in Scotland, Wales and Northern Ireland supported the formation of an integrated clinical academic training (IAT) pathway[6] designed to tackle concerns about dwindling clinical academic numbers by providing clinical training posts with protected research time. In England, clinical academic training opportunities start with Academic Foundation posts, continuing

to predoctoral academic clinical fellowships (ACFs), doctoral training fellowships and, finally, postdoctoral clinical lectureships (CLs).[6] Despite these efforts, the latest report from the Medical Schools Council[7] states that more than half of UK medical schools report difficulties in recruitment to some clinical academic posts. Of additional concern is the small proportion of women who pursue clinical academic careers (CACs).[8]

A pivotal point common to clinical academic training pathways irrespective of country is the PhD.[6] Interestingly, recent career tracking of doctoral clinical training fellowships awarded by major UK research funders[9] suggests that only around a third of holders progress to a junior postdoctoral CL or the more senior clinician scientist post. (CLs are open to doctors in specialty training, whereas clinician scientists have completed their training and developed research independence.) A range of barriers and enablers have been identified as contributing to this issue, including matters relating to work-life balance, funding, mentorship and careers information.[9–12] Notwithstanding, there has been no work in the UK examining the characteristics of doctors undertaking PhDs or how they view their future career options, including why those wanting a CAC may not consider applying for a postdoctoral CL.[10 13]

In 2013, the Oxford University Clinical Academic Graduate School (OUCAGS) began a longitudinal study of its clinical PhD students to help understand these issues. The part of the study reported here was undertaken in 2015–2016 in collaboration with University College London's (UCL) Academic Careers Office (ACO). Our aims were to: describe the demographic and clinical-training characteristics of both institutions' clinical PhD students; determine their long-term career plans (including how many want a CAC) and factors influencing these plans; examine their views regarding CL posts, since these are the first postdoctoral posts on the established IAT pathway, and explore their preparedness for making career decisions.

## METHOD
In reporting our study, we followed the STROBE guidelines.[14]

### Design and procedures
We conducted a census survey of all medically qualified PhD students enrolled, at the beginning of autumn 2015, at two universities in England: the University of Oxford (hereafter 'Oxford') and UCL. Students who were either suspended or had already submitted their thesis were excluded. MD/DM students were also excluded as their higher degree programme is often less structured and the survey questions may not apply in the same way. Each survey had ethical approval from its home institution and collected its data via its own online questionnaire between the end of October 2015 and the beginning of March 2016. Participants received: an email from senior

academics introducing the study; an email invitation which offered an £8 Amazon voucher for participation; two email reminders at Oxford, three at UCL, and one postal reminder inviting them to complete either the online questionnaire or a paper version. The target population was identified from administrative records. Where we were unclear about students' qualifications, after the introductory email we sent an email checking qualifications and excluded those who responded that they were not medically qualified. A total of 523 eligible students were asked to participate (see STROBE diagrams in online supplementary box 1).

### Instruments
OUCAGS developed the questionnaire in 2013, as part of its Clinical DPhil Paths longitudinal study (2013–2018) of the career plans of clinical PhD students. ('DPhil' is the term for an Oxford PhD.) Questions were either newly created or adapted from existing questionnaires.[15 16] They covered respondents' characteristics and career plans, including: demographic variables, medical training, predoctoral involvement in research, reason for entering the doctorate, doctoral experience, intention to pursue a CAC and a CL in England (where the NIHR funds a large number of posts) and views regarding CL posts (including how attainable they think these posts are[17]) (see online supplementary box 2 for details of questionnaire development, the questions reported below and minor adaptations by UCL (eg, use of 'PhD' instead of 'DPhil')).

### Analysis
We merged anonymised Oxford and UCL datasets and analysed the data using SPSS V.22. We used frequencies to profile respondents (eg, age and fee status) as well as to compare respondents by gender and intention towards a CAC. For fee status, we grouped UK and EU respondents together as both were eligible for 'home' fees and have equal employment rights within the UK. This also preserved EU respondents' anonymity in file sharing as their number was small (n=12 and 9 respondents from Oxford and UCL, respectively).

We analysed responses to open questions using conventional content analysis[18] to identify themes and counted the number of participants mentioning each recurrent theme. Responses were double-coded by JL and VR in a face-to-face meeting, and any coding inconsistencies were discussed and resolved jointly (see online supplementary box 3).

As a first step, the data were examined in detail. Then variables with data collected on five-point or six-point scales were summarised into three categories (eg, 'a great deal/a lot', 'a moderate amount' and 'a little/none at all/not applicable'—see Results tables), with the full set of response categories referred to where relevant (eg, for very skewed variables). Using three categories meant that data could be more clearly presented and, in most cases, items from a same question could be entered into $\chi^2$ tests

with responses categorised in the same manner. Categories were further collapsed, where necessary, to meet the requirements of $\chi^2$ tests for expected frequencies.[19]

We conducted $\chi^2$ tests of association to determine which of a range of background, doctoral experience and attitude variables were associated (at the 0.05 level of statistical significance) with two key career outcomes:

1. intention to pursue a CAC in the long term (respondents saying CAC vs those with other plans or undecided) and
2. likelihood of seeking a CL in England (extremely or very likely to seek such a post vs not at all to moderately likely).

For statistically significant $\chi^2$ tests, we referred to adjusted standardised residuals to assist with interpreting results. The higher a residual's value, the more a cell contributes to the $\chi^2$ value.

## RESULTS
### Characteristics of clinical PhD students
#### Response rates
Response rates were 67% and 57% for the Oxford and the UCL surveys, respectively, giving 322 respondents in total. A more resource-intensive approach (eg, telephone reminders/interviews) may have yielded higher response rates.[20] Only two Overseas (ie, non-UK/EU) students responded to the UCL questionnaire and, to preserve their anonymity, their data were not shared with Oxford. This paper therefore reports on data from 320 clinical PhD students, of whom 170 (53%) were enrolled at Oxford and 150 (47%) at UCL.

#### Demographics and training
Of the 320 respondents, 54% (n=174) were male, 44% (n=142) female and 1% (n=4) did not specify. Analysis for non-response bias by gender showed no bias needed to be accounted for in the analyses. Indeed, respondents were representative of the combined populations of clinical PhD students of Oxford and UCL as, based on administrative data, 57% (n=299) of students invited to complete the questionnaire were male and 43% (n=224) were female.

Most respondents were aged 30–35 years (table 1). Figures for UK/EU respondents only (N=279) are comparable to figures for all respondents, as 52% (n=146) were male and only 10% (n=28) were under 30 years. However, among the Overseas (Oxford) students (N=33), more respondents were male (70%, n=23) and under 30 years (49%, n=16) (see online supplementary box 4 for additional information).

Most participants were in specialty training years 3–8 (ST3–8), that is, had done four or more years of clinical training after medical school and entered a specialty; this included 80 respondents (25% of the total) in years 6–8 (ST6–8), who were close to completing training. Approximately one-fifth had completed their training (table 1).

The majority (84%, n=268) of participants had been involved in medical research prior to their doctorate, but over one-tenth (12%, n=38) had no involvement (see online supplementary box 5 for additional information). This was similar at Oxford and UCL. The most common type of predoctoral medical research experience was an intercalated degree, that is, a research degree taken during the undergraduate years (48% of respondents, n=154). Also, over one-fifth (21%, n=67) of participants had held a predoctoral fellowship (ie, an ACF in England, Wales or Northern Ireland (n=65) or a clinical research fellowship (CRF) in Scotland (n=2)). In addition, some respondents had completed an Academic Foundation Programme (19%, n=60), held other posts with time for research (12%, n=37) and/or had 'other' research-related experiences, such as research towards an MSc thesis or a research assistantship (31%, n=100).

#### Reason for entering a doctoral programme
Respondents' main reason for undertaking a doctorate centred on interest in their area of research (46%, n=146) or improving their prospects for an academic/ research career (42%, n=134). However, improving their prospects for a clinical career *outside* academia was also mentioned by some (10%, n=33) (see online supplementary box 5 for additional information). Respondents who had completed their specialty training were particularly likely to cite interest in their area of research (56%, n=32), although over one-third (35%, n=20) specifically wanted to improve their prospects for an academic/ research career.

### Long-term career planning
#### Career plans
Half of all respondents expressed a desire to pursue a CAC in the long term, and around a third intended to work in clinical posts with some research, or some teaching and research (table 1). The figure for respondents wanting a CAC was higher among former ACFs/CRFs (63%, n=42). Over one-tenth of respondents, including 13% (n=9) of former ACFs/CRFs and 12% (n=27) of doctors in specialty training, were undecided about their long-term career, although very few wanted purely clinical posts or clinical posts with teaching. The proportion of participants undecided or with each type of career plan was similar across doctoral year groups (year 1 to year 4 and above).

$\chi^2$ tests showed that greater predoctoral involvement in medical research, and pursuing a doctorate mainly to improve academic/research-career prospects, were each associated with wanting a CAC (table 2). In contrast, there was no statistically significant association between whether a respondent was planning a CAC and gender, fee status (UK/EU vs Overseas), age or training stage (see online supplementary box 6).

Enthusiasm and commitment to a given career path, as well as wanting intellectual stimulation and to make a difference to patients, were major influencing factors in respondents' long-term career planning (table 3 and

**Table 1**  Respondents' age, training level and long-term career plans: percentages by gender and for all respondents

| | All respondents* | | |
| --- | --- | --- | --- |
| | Males, % (N=174) | Females, % (N=142) | Total, % (N=320) |
| **Age, years** | | | |
| <30 | 17 | 11 | 14 |
| 30–35 | 56 | 54 | 55 |
| 36+ | 14 | 16 | 15 |
| No response | 13 | 20 | 17 |
| **Highest training stage completed or underway†** | | | |
| Pre-Foundation, in Foundation or post-Foundation programme | 14 | 10 | 12 |
| In ST1–2, CT or ACCS *or* completed ST1–2, CT or ACCS | 12 | 8 | 10 |
| In ST3–8 | 56 | 57 | 56 |
| ST level not specified | 3 | 4 | 4 |
| Completed specialty training | 16 | 21 | 18 |
| **Long-term career plans** | | | |
| Clinical academic posts | 55 | 44 | 50 |
| Clinical service posts with some teaching and research | 23 | 23 | 23 |
| Clinical service posts with some research | 9 | 9 | 9 |
| Clinical service posts with some teaching | 2 | 1 | 2 |
| Clinical service posts *without* teaching or research | 1 | 1 | 1 |
| Research-only posts | 2 | 3 | 3 |
| Undecided | 8 | 15 | 11 |
| Other | 1 | 4 | 2 |
| No response | – | 1 | <1 |

Due to rounding, column percentages may not add to 100.
*Four respondents did not provide their gender.
†In the UK, following medical school and a 2-year Foundation programme, doctors enter ST. The first 2 years of ST are usually referred to as ST1–2, but some specialties have, before ST, a CT or ACCS programme. However, this is highly dependent on specialty.
ACCS, acute care common stem; CT, core training; ST, specialty training.

see online supplementary box 7 for further details). In addition, having acceptable working hours and conditions was especially important to respondents who were undecided about their long-term career, with 78% (n=28) of this group saying this had a great deal or a lot of importance.

In $\chi^2$ tests, the following were positively associated with wanting a CAC (table 2): wanting intellectual stimulation, enthusiasm/commitment, self-appraisal of own skills/aptitudes and career promotion and prospects. Also, a perceived lack of posts in other possible career paths was negatively associated with planning a CAC, possibly as respondents wanting a CAC can always return to a full-time clinical career. There was no statistically significant association between planning a CAC and other long-term career planning factors examined (see online supplementary box 6).

### Drawbacks of a clinical academic career
Respondents not intending to work in either clinical academia or research-only posts (N=153) were asked how much each of a range of factors might be a reason for excluding a CAC from their long-term career plans. This group included over a third (37%, n=25) of former ACFs/CRFs. Nearly two-thirds of the group (66%, n=101) said that the difficulty of obtaining research grants had a great deal or a lot of weight in their planning. Other factors with considerable weight were (% saying 'a great deal' or 'a lot'):
► the small number of senior academic appointments available (50%, n=76),
► the competing pressures from service, teaching and research (47%, n=72),
► the limited future financial prospects compared with alternative careers (34%, n=52),
► needing work arrangements that are compatible with their caring responsibilities (30%, n=46, including 45%, n=34, of women but only 15%, n=11, of men) and
► family circumstances making them unwilling to move for work (27%, n=42).

Responses to the open question 'what might make a CAC more attractive to you?', asked of all participants,

**Table 2** Variables associated with wanting a CAC versus having other plans or being undecided: statistically significant $\chi^2$ tests

| Variables | Groupings | Respondents wanting a CAC | | | $\chi^2$ test | | |
| | | % | n / N | Adjusted residuals* | df | $\chi^2$ | p Value |
|---|---|---|---|---|---|---|---|
| Predoctoral involvement in medical research | No involvement | 26 | 10/38 | –3.0 | 2 | 22.193 | 0.0005 |
| | Some involvement (intercalated degree <u>or</u> post(s) with research, possibly with other experience, or 'other' experience)† | 46 | 95/205 | –1.4 | | | |
| | High involvement level (intercalated degree <u>and</u> post(s) with research and possibly 'other' experience, too)† | 73 | 45/62 | 4.1 | | | |
| Main reason for doctorate | Interest in my area of research | 49 | 71/146 | –0.4 | 2 | 25.515 | 0.0005 |
| | Improving prospects for an academic/research career | 61 | 82/134 | 3.5 | | | |
| | Other | 15 | 6/39 | –4.6 | | | |
| **Influences on long-term career thinking:** | | | | | | | |
| Wanting intellectual stimulation‡ | A great deal/A lot | 55 | 157/284 | 5.4 | 1 | 29.640 | 0.0005 |
| | A moderate amount/A little/Not at all | 6 | 2/34 | –5.4 | | | |
| Enthusiasm/commitment‡ | A great deal/A lot | 52 | 156/300 | 2.9 | 1 | 8.480 | 0.004 |
| | A moderate amount/A little/Not at all | 17 | 3/18 | –2.9 | | | |
| Self-appraisal of own skills/aptitudes | A great deal/A lot | 56 | 107/191 | 2.6 | 2 | 7.743 | 0.021 |
| | A moderate amount | 39 | 38/98 | –2.7 | | | |
| | A little/Not at all | 48 | 14/29 | –0.2 | | | |
| Career promotion and prospects | A great deal/A lot | 58 | 85/146 | 2.7 | 2 | 7.300 | 0.026 |
| | A moderate amount | 43 | 51/118 | –1.9 | | | |
| | A little/Not at all | 43 | 23/54 | –1.2 | | | |
| Lack of posts in other possible career paths | A great deal/A lot | 27 | 8/30 | –2.7 | 2 | 14.357 | 0.001 |
| | A moderate amount | 36 | 19/53 | –2.3 | | | |
| | A little/Not at all | 56 | 132/235 | 3.7 | | | |
| **CL posts:** | | | | | | | |
| CL knowledge | A great deal/A lot | 73 | 27/37 | 3.0 | 2 | 19.198 | 0.0005 |
| | A moderate amount | 59 | 69/118 | 2.3 | | | |
| | A little/Nothing at all | 39 | 63/162 | –4.1 | | | |
| CL attainability (accuracy of statement that CL is attainable) | Extremely/Very accurate | 63 | 60/96 | 2.9 | 2 | 16.079 | 0.0005 |
| | Moderately accurate | 54 | 57/105 | 1.1 | | | |
| | Slightly/Not at all accurate | 36 | 42/117 | –3.8 | | | |
| **Sources of information about possible career paths (amount of information received):** | | | | | | | |
| OUCAGS/ACO | A great deal/A lot | 82 | 22/27 | 3.4 | 2 | 18.917 | 0.0005 |
| | A moderate amount | 65 | 33/51 | 2.2 | | | |
| | A little/None at all/N/A | 44 | 103/236 | –4.1 | | | |
| Mentor(s) | A great deal/A lot | 64 | 50/78 | 2.8 | 2 | 8.778 | 0.012 |
| | A moderate amount | 49 | 40/81 | –0.2 | | | |
| | A little/None at all/N/A | 44 | 68/156 | –2.3 | | | |

*For explanation and interpretation see Method section.

†The following were considered 'posts with research': Academic Foundation Programme posts, Academic Clinical Fellow (England, Wales, Northern Ireland), Clinical Research Fellow (Scotland) and responses of 'other post with time for research'.

‡Due to the small numbers selecting 'A moderate amount', 'A little' and 'Not at all', these categories were merged in order to meet $\chi^2$ test assumptions (see Method section).

ACO, Academic Careers Office; CAC, clinical academic career; CL, clinical lectureship; OUCAGS, Oxford University Clinical Academic Graduate School.

**Table 3** Factors influencing respondents' thinking about their long-term career: percentage saying 'a great deal' or 'a lot', by whether respondents want a CAC versus having other plans or being undecided and for all respondents

| | Respondents wanting a CAC*, % (N=159) | Respondents not wanting a CAC*, % (N=160) | Total, % (N=320) |
|---|---|---|---|
| Enthusiasm/commitment (A great deal)† | 98 (67) | 90 (52) | 94 (60) |
| Wanting intellectual stimulation (A great deal)† | 99 (63) | 79 (38) | 89 (50) |
| Wanting to make a difference to patients (A great deal)† | 88 (51) | 79 (41) | 84 (46) |
| Self-appraisal of own skills/aptitudes | 67 | 53 | 60 |
| Wanting a career that fits my domestic circumstances | 56 | 54 | 55 |
| Wanting a career with acceptable working hours/conditions | 50 | 59 | 54 |
| Exposure to role models | 54 | 44 | 49 |
| Career promotion and prospects | 54 | 38 | 46 |
| Future financial prospects | 39 | 38 | 38 |
| Advice from others | 25 | 19 | 22 |
| Lack of posts in other possible career paths | 5 | 14 | 9 |

*Base: all participants (N=319) who responded to the question about their career plans.
†Some variables were considerably skewed. Where the mode for the whole group of respondents (total) was 'a great deal', figures for the modal response-only are provided in brackets.
CAC, clinical academic career.

confirmed that issues around obtaining research grants, securing posts and career progression are of particular concern when considering a CAC, as responses centred around these themes (see online supplementary box 3).

### Clinical lectureships

Over three-fifths of participants (table 4), including 65% (n=181) of UK/EU respondents, were at least moderately likely to seek a CL post in England at an appropriate point after completing their doctoral studies. This figure rose to 78% (n=52) for former ACFs/CRFs and to 98% (n=41) among former ACFs/CRFs wanting a CAC. In addition, nearly two-thirds of all participants and over four-fifths of former ACFs/CRFs (82%, n=55) perceived CL posts to be at least moderately attainable for them (table 4). Although overall women were more cautious about the attainability of CL posts, among respondents wanting a CAC similar proportions of men (74%, n=71) and women (73%, n=46) felt they were at least moderately attainable.

Respondents in specialty training, and those aged 30–35, were the most likely to seek a CL post in England. This is supported by $\chi^2$ tests (table 5), which also revealed positive associations between likelihood of seeking a CL in England and wanting a CAC, being from the UK/EU, predoctoral involvement in medical research, knowledge of CLs and feeling that a CL is attainable. However, there was no statistically significant association between likelihood of seeking a CL post and gender (see online supplementary box 8). $\chi^2$ tests also showed a positive association between wanting a CAC and perceiving that CL posts are attainable (table 2).

Although many respondents thought that they may successfully apply for a CL, only a minority (table 4) felt that they knew a great deal or a lot about CL posts in England. In fact, just over half, including one-quarter (25%, n=17) of former ACFs/CRFs, felt that they had little or no knowledge (table 4). Also, although $\chi^2$ tests showed a positive association between wanting a CAC and knowledge of CL posts (table 2), two-fifths (40%, n=63) of those planning a CAC nevertheless had little or knowledge of these CL posts.

### Career preparedness

Most respondents (64%, n=204) felt that their doctoral programme was preparing them extremely or very well for a possible future CAC, although this figure was higher for men (72%, n=125) than for women (55%, n=78).

A proportion (23%, n=75) of respondents had little or no encouragement from their PhD supervisor(s) to reflect on their career development needs, but most (56%, n=180) had a formal or informal mentor they met regularly. However, only 50% (n=71) of women had such a mentor, compared with 62% (n=107) of men. Also, over a third (34%, n=109) of all respondents stated that they did not have a mentor but would like to find one, and this figure increased to 41% (n=58) among women (vs 29%, n=50, among men).

Respondents who wanted a CAC were more likely to say that they get a great deal or a lot of information about possible careers from mentors (31%, n=50 vs 18%, n=28, for those who had other plans or were undecided). In contrast, over three-fifths of women *not* intending to

**Table 4** Respondents' likelihood of seeking a CL, and views on CL attainability and own knowledge about CLs: percentages by gender and for all respondents

| | All respondents* | | |
| --- | --- | --- | --- |
| | Males (%) | Females (%) | Total (%) |
| Likelihood of seeking a CL† | N=173† | N=141† | N=318† |
| Extremely/Very likely | 45 | 40 | 42 |
| Moderately likely | 17 | 23 | 20 |
| Slightly/Not at all likely | 36 | 36 | 36 |
| No response | 1 | 2 | 2 |
| CL attainability accuracy of statement that respondent could become a CL in England, if wanted | N=174 | N=142 | N=320 |
| Extremely/Very accurate | 37 | 22 | 30 |
| Moderately accurate | 29 | 36 | 33 |
| Slightly/Not at all accurate | 33 | 42 | 37 |
| No response | 1 | – | <1 |
| Knowledge of CL amount known | N=174 | N=142 | N=320 |
| A great deal/A lot | 12 | 11 | 12 |
| A moderate amount | 40 | 34 | 37 |
| A little/Nothing at all | 48 | 54 | 51 |
| No response | – | 1 | 1 |

Due to rounding, column percentages may not add to 100.

*Four respondents did not provide their gender.

†The question about likelihood of seeking a CL was not asked of two participants who stated that they intended to work in clinical posts without teaching or research.

CL, clinical lectureship.

pursue a CAC received little or no such information from senior academics (64%, n=50 vs 50%, n=39, of men) and mentors (63%, n=49 vs 45%, n=35, of men) (online supplementary box 9 contains additional data).

Men and women obtained similar amounts of information from OUCAGS/UCL's ACO, with 28% (n=48) of men and 22% (n=31) of women receiving at least moderate amounts of information from this source. Moreover, receiving information about careers from OUCAGS/ACO and mentor(s) each had a positive relationship with both wanting a CAC and likelihood of seeking a CL (tables 2 and 5). The latter also was positively associated with receiving information about careers from academic/research supervisor(s) and senior academics. Interestingly, there were no statistically significant relationships between either wanting a CAC or likelihood of seeking a CL and having a mentor, encouragement from supervisors to reflect on career needs or information from peers and newsletters (see online supplementary boxes 6 and 9).

## DISCUSSION

The clinical PhD students in our study are aged mainly between 30 and 35 years, in specialty training (with a quarter close to completion) and gender balanced once Overseas students are accounted for. Encouragingly, half of our clinical PhD students want to pursue a CAC, with women as likely as men to express this career goal. Among those who have already completed their clinical specialty training (nearly one-fifth of our clinical PhD students), there is also interest in CACs. Indeed, training level does not seem to affect long-term plans and neither does age. Greater predoctoral involvement in medical research is strongly associated with wanting a CAC but, unexpectedly, less than two-thirds of former ACFs/CRFs, that is, PhD students already on the clinical academic path, are planning to pursue a CAC. Given that our cohort, who secured doctorates at Oxford/UCL, might be regarded as 'high achievers', it is surprising that a higher proportion did not want a CAC. On a more positive note, many PhD students, even if not planning a CAC, *are* interested in a career that combines clinical work and research, with very few wanting purely clinical posts. In common with others,[9–11] we found that uncertainties around funding and career progression influence career plans and are perceived as drawbacks of CACs. Respondents not planning a CAC, including former ACFs/CRFs, are particularly worried about the small number of senior academic appointments available as well as the difficulty of obtaining research grants and work-life balance. While

**Table 5** Variables associated with likelihood of seeking a CL: statistically significant $\chi^2$ tests

| Variables | Groupings | Respondents extremely/very likely to seek a CL | | | $\chi^2$ test | | |
| | | % | n/N | Adjusted residuals* | df | $\chi^2$ | p Value |
|---|---|---|---|---|---|---|---|
| Fee status | UK/EU | 47 | 128/273 | 3.4 | 1 | 11.383 | 0.001 |
| | Non-EU (Overseas) | 16 | 5/32 | −3.4 | | | |
| Age | <30 years | 27 | 12/44 | −2.5 | 2 | 11.534 | 0.003 |
| | 30–35 years | 51 | 89/173 | 3.4 | | | |
| | 36 and over | 32 | 14/44 | −1.8 | | | |
| Training stage | Pre-specialty | 22 | 8/36 | −2.7 | 2 | 12.149 | 0.002 |
| | In specialty | 49 | 108/221 | 3.4 | | | |
| | Completed | 32 | 18/56 | −1.8 | | | |
| Predoctoral involvement in medical research | No involvement | 21 | 8/38 | −2.9 | 2 | 24.502 | 0.0005 |
| | Some involvement (intercalated degree <u>or</u> post(s) with research, possibly with other experience, or 'other' experience)† | 39 | 77/199 | −1.9 | | | |
| | High involvement level (intercalated degree <u>and</u> post(s) with research and possibly 'other' experience, too)† | 68 | 42/62 | 4.5 | | | |
| Long-term career plans | CAC | 67 | 105/158 | 8.5 | 3 | 72.557 | 0.0005 |
| | Clinical service with research (possibly also teaching) | 18 | 18/99 | −6.0 | | | |
| | Undecided | 23 | 8/35 | −2.5 | | | |
| | Other plans | 15 | 3/20 | −2.6 | | | |
| CL knowledge | A great deal/ A lot | 65 | 24/37 | 2.9 | 2 | 31.140 | 0.0005 |
| | A moderate amount | 57 | 66/116 | 3.8 | | | |
| | A little/Nothing at all | 28 | 44/158 | −5.5 | | | |
| CL attainability (accuracy of statement that CL is attainable) | Extremely/Very accurate | 70 | 66/95 | 6.3 | 2 | 52.138 | 0.0005 |
| | Moderately accurate | 44 | 46/105 | 0.2 | | | |
| | Slightly/Not at all accurate | 20 | 22/112 | −6.2 | | | |
| **Sources of information about possible career paths (amount of information received):** | | | | | | | |
| Academic/ research supervisor(s) | A great deal/A lot | 48 | 49/103 | 1.2 | 2 | 6.443 | 0.040 |
| | A moderate amount | 48 | 52/108 | 1.3 | | | |
| | A little/None at all/N/A | 33 | 33/101 | −2.5 | | | |
| Senior academics | A great deal/A lot | 50 | 31/62 | 1.3 | 2 | 9.356 | 0.009 |
| | A moderate amount | 53 | 48/91 | 2.2 | | | |
| | A little/None at all/N/A | 35 | 55/159 | −3.0 | | | |
| OUCAGS/ACO | A great deal/A lot | 59 | 16/27 | 1.8 | 2 | 17.804 | 0.0005 |
| | A moderate amount | 65 | 33/51 | 3.5 | | | |
| | A little/None at all/N/A | 36 | 82/230 | −4.2 | | | |
| Mentor(s) | A great deal/A lot | 57 | 43/76 | 2.7 | 2 | 10.053 | 0.007 |
| | A moderate amount | 46 | 37/81 | 0.6 | | | |
| | A little/None at all/N/A | 35 | 53/152 | −2.9 | | | |

*For explanation and interpretation, see Method section.
†The following were considered 'posts with research': Academic Foundation Programme posts, Academic Clinical Fellow (England, Wales, Northern Ireland), Clinical Research Fellow (Scotland) and responses of 'other post with time for research'.
ACO, Academic Careers Office; CAC, clinical academic career; CL, clinical lectureship; OUCAGS, Oxford University Clinical Academic Graduate School.

the desire for a career that fits domestic circumstances is not statistically associated with career plans, these issues are given more weight by women than men. Such factors may go on to influence women's actual postdoctoral career choices,[21] as women may evaluate barriers encountered differently to men, leading to a change in their career intentions.[22]

We were curious about the place of CL posts in career plans. These posts might be considered a logical step in a postdoctoral CAC but a recent report[9] suggests their take-up is lower than expected. In light of this, our data are encouraging as around three-fifths of respondents were at least moderately likely to apply for a CL post—a greater proportion than were planning a CAC. This might indicate that some view CL posts as valuable for career progression, whether as part of a CAC or not. Concomitantly, however, progression to a CL may be affected by the low levels of knowledge about the posts. Additionally, we found some issues with career preparedness (also reported by others[9]). These were more prominent among women, who were more likely not to have a mentor but want one and, if not wanting a CAC, received less guidance from senior academics. Exposure to role models, often suggested as important[23 24] in career development, did not appear to be significant among our respondents, but the positive association between receiving information from OUCAGS/ACO and wanting a CAC is reassuring as it supports the benefit of such infrastructure.

### Strengths and limitations of the study

Retention of postdoctoral clinical academics is crucial to the progress of academic medicine. However, the relevant literature is predominantly North American[8] and also tends to rely on retrospective data.[9] Ours is the first UK study to describe, based on data from two universities, the career aspirations of medically qualified PhD students and what influences them as they make their career plans. As it is a cross-sectional study, causality cannot be inferred from the associations we found between career plans and other variables. It is also not possible to say whether respondents' intentions will be implemented as planned and our findings may not generalise to other institutions as our data are from two research-intensive universities in England.

### Implications of findings

Our study, which adds the perspective of UK clinical PhD students to the existing literature, suggests four areas of action that will support the development of a stable pipeline of clinical academics. First, we suggest promoting increased numbers of flexible opportunities for doctors to gain medical research experience, starting from their undergraduate studies. Indeed, we found that those with the most research experience were more likely to want a CAC and, although research experience may not directly influence that choice, it may contribute to an informed career decision. Second,

institutions should offer comprehensive career support for their clinical doctoral students. This could be via delivery of a targeted careers package by a local institutional unit similar to OUCAGS/ACO which could address knowledge about CL posts and other postdoctoral opportunities; facilitate mentors and doctoral students to develop individualised career plans; meet the unfulfilled need for mentors; brief senior academics on current CAC opportunities to enable them to provide up-to-date guidance. Third, we suggest a need to increase the flexibility of CAC paths even further than at present. CL posts are the first postdoctoral position to afford clinicians the opportunity to develop as independent research leaders and, without this opportunity, progressing further academically may be difficult. We have identified that family mobility is an issue for our clinical PhD students, but identifying a CL post in the right specialty, in the right place and at the right time (eg, as the doctorate comes to an end) is challenging and sometimes impossible. This is exacerbated by eligibility criteria, particularly for NIHR CL posts, requiring that applicants have at least 1 year of training remaining (the post ceases at completion of training), since, as we found, many clinical doctoral students have completed, or nearly completed, their training. We therefore suggest that criteria for CL posts be relaxed. Various postdoctoral points of entry into CL posts could be allowed (including after training completion) and the posts should not be specialty-specific so as to afford enhanced opportunities to all postdoctoral clinicians. Increased flexibility might also attract back postdoctoral clinicians who had originally not pursued a CAC, such as when family circumstances change and work-life balance may seem more achievable. Such changes could help medical schools overcome the recruitment challenges that have been reported.[7] Finally, more senior posts are required since their lack seems to significantly influence career decisions. Universities are unlikely to be able to create sufficient fully funded senior posts, so formalising research time in a subset of NHS consultant posts, and establishing them as joint university appointments, could cost-effectively expand the definition of a CAC. Ensuring the visibility of these posts would demonstrate a clear career trajectory to encourage postdoctoral retention. Although current NHS service and funding pressures might make this seem low priority, we would argue that such joint appointments are essential as research activity is centrally positioned in the NHS constitution[25] and, crucially, it improves patient care and service quality.[26 27]

### Conclusion and further research

Our findings suggest that many doctoral students with the potential to be the clinical academics of the future may never tread this path, some (former ACFs/CRFs) may choose to abandon it, and some may be excluded by inflexible eligibility criteria for postdoctoral posts. It is therefore important for those promoting

CACs to provide sufficient, flexible opportunities to engage in clinical academia to all postdoctoral clinicians. This should include tackling issues around the career pathway beyond the clinical training period and building career preparedness and resilience. Our findings also suggest that an additional approach to increasing the numbers of clinical academics might be to ensure predoctoral doctors have ample opportunities to gain research experience. Further research should investigate whether clinical PhD students' intentions will be implemented as planned (eg, will equal numbers of men and women actually develop CACs?) and seek to identify which factors contribute to actually pursing a CAC. The Clinical DPhil Paths longitudinal study is currently collecting data on this in the UK.

**Acknowledgements** We thank the clinical PhD students who took part in our study and Abigail Hipkin (Research Education and Training Administrator, OUCAGS), who assisted with data collection at Oxford. We are also grateful to the University of Oxford and UCL administrative services and the UCL graduate tutors who helped us identify the students to invite to participate in our study.

**Contributors** JL, DB and CP developed the Clinical DPhil Paths questionnaire. JL collected the Oxford data and VR collected the UCL data. All authors conceived the data analysis plan and later made substantive contributions to the interpretation of the findings and the writing of this article. JL conducted the quantitative data analysis and reporting, with assistance from VR and expert statistical guidance from TL. JL and VR undertook the qualitative data analysis. JL and DB wrote the first and subsequent drafts. All authors critically reviewed and edited drafts and approved the final version of the manuscript. They also had full access to all of the data (including statistical reports and tables) in the study and can take responsibility for the integrity of the data and the accuracy of the data analysis. The study guarantors are DB (for Oxford) and GR (for UCL).

**Funding** Oxford—This research was funded by OUCAGS, which is part of the University of Oxford Medical School. DB is, in part, supported by the National Institute for Health Research (NIHR) Oxford Biomedical Research Centre (BRC). UCL—This research was funded by the National Institute for Health Research University College London Hospitals Biomedical Research Centre. NJF and HB were (in part) supported by the National Institute for Health Research (NIHR) Collaboration for Leadership in Applied Health Research and Care (CLAHRC) North Thames at Bart's Health NHS Trust. The views expressed are those of the author(s) and not necessarily those of the NHS, the NIHR or the Department of Health. The authors conducted the study with no involvement from the funders.

**Competing interests** All authors have completed the ICMJE uniform disclosure form at . JL, CP and DB declare that JL and DB are employed by, and CP directs, OUCAGS, which exists to promote and advance clinical academic careers; CP declares that his salary is, in part, paid for by Health Education England in the Thames Valley; VR and GR declare that the former works for, and the latter manages, UCL's Academic Careers Office, which is funded by UCLH's NIHR Biomedical Research Centre, and thus the National Institute for Health Research; no other financial relationships with any organisations that might have an interest in the submitted work in the previous three years besides the support provided for the submitted work by the organisations described above under 'funding'; no other relationships or activities that could appear to have influenced the submitted work.

**Ethics approval** Oxford survey approved by the University of Oxford's Medical Sciences Inter-Divisional Research Ethics Committee (reference MSD-IDREC-C1-2014-149). UCL survey approved by the UCL Research Ethics Committee (application 7335/002).

**Provenance and peer review** Not commissioned; externally peer reviewed.

**Data sharing statement** No additional data are available.

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
