## [Reviewer comments · BMJ Open]

ARTICLE DETAILS

TITLE (PROVISIONAL)	The clinical academic workforce of the future: a cross-sectional study of factors influencing career decision-making amongst clinical PhD students at two research-intensive UK universities
AUTHORS	Lopes, Joana; Ranieri, Veronica; Lambert, Trevor; Pugh, Chris; Barratt, Helen; Fulop, Naomi; Rees, Geraint; Best, Denise

VERSION 1 - REVIEW

REVIEWER	Dr Simon Tso Jephson Dermatology Center, South Warwickshire NHS Foundation Trust, United Kingdom I have previously published in the same field. Tso S, Potential barriers to pursuing a career in academic dermatology. 2016. 175(1):222-223. DOI: 10.1111/bjd.14519 Tso S. Clinical academic career: an alternative viewpoint. Clinical Teacher. (in press) doi: 10.1111/tct.12545
REVIEW RETURNED	16-Mar-2017

GENERAL COMMENTS	I read the manuscript with great interest which covers an important topic. However, prior to acceptance for publication, I would recommend the authors to consider my comments and recommendations on how this piece of work could be strengthened. 1) Statistics: please review all statistical data to ensure the % will add up to 100%, for example Page 9 Table 1 line 10-15 female respondents % added up to 101%. In other tables, the statistics added up to 99% or 101%. 2) Page 35 Box 5. Please provide justification for classifying specialities into craft and non-craft specialties. For example, the specialty I come from, Dermatology, many dermatology surgeons undertake Moh's micrographic surgery and undertake surgical fellowships. They would consider their work as a craft as well. Likewise, interventional cardiology colleagues may also feel the same way too. Otherwise, Box 5 (and all relevant results) should be presented in an alternate format. 3) Qualitative analysis: There is insufficient information about the qualitative analysis method employed in this study to permit interpretation of the data. Looking at the results, I believe you were using content analysis to count the frequency of responses with similar codes/themes. The method used should be referenced where appropriate. How did the two universities compare the codes (face-to-face vs teleconference)? 4) Study population: Were MD (research) students excluded from this study? If so, why? Did the study include suspended students?
--

	Were part time students included? 5) Clinical academic career: Could the authors comment in the methods section on why the study narrowly chosen the role of 'Clinical Lecturership' as an end point when there are so many different clinical academic roles out there? You may wish to comment on the lack of clear career path in clinical academia 6) Page 4 line 16 please change the word 'fears' to an alternative, for example, 'concerns' 7) Methods: page 5: how much time were given between the two / three email reminders? When was the online or paper questionnaire sent out? 8) Page 4 line 40 Ref 9-11. You may wish to read the work by Bucklin et al 2014, Lyons et al 2010 and Alenxander and Lang (2008) which may help you to expand on the range of barriers and enablers contributing to the issue under investigation. Bucklin BA, Valley M, Welch C, Tran ZV, Lowenstein SR. Predictors of early faculty attrition at one Academic Medical Center. BMC Med Educ 2014;14:27. Lyons OTA, Smith C, Winston JS, Geranmayeh F, Behjati S, Kingston O, Pollara G. Impact of UK academic foundation programmes on aspirations to pursue a career in academia. Med Educ 2010;44(10):996–1005. Alexander H, Lang J. The long-term retention and attrition of U.S. Medical School Faculty. Association of American Medical Colleges 2008;8:4.
--	--

REVIEWER	Danielle Rodin University of Toronto, Canada
REVIEW RETURNED	15-Apr-2017

GENERAL COMMENTS	This paper presents findings from a cross-sectional study of the factors that contribute to the career intentions of 320 medically qualified doctoral students at two research-intensive UK universities. This study was undertaken because of concerns about the retention of such trainees in academic medicine. It is the first study of its kind in the UK context and the first cross-sectional study on this topic. Based on a large sample size, the investigators found that the perception of limited opportunities and unsatisfactory work-life balance contributed to career intentions and that only half of the surveyed intended to pursue an academic career. Lack of information and exposure to opportunities affected outcomes, although it is notable that gender did not affect the outcomes assessed. A number of recommendations are made to increase the likelihood of an academic career intention. Issues that the authors might wish to address include:  1. The authors include the questionnaire in the Appendix. However, it would be helpful if information was provided about how the questionnaire was developed, whether items were drawn from existing questionnaires or from the literature, or whether the questionnaire has been validated in any way. 2. The study is predominantly quantitative, but the questionnaire includes several open-ended questions, from which responses were
--

	analyzed by thematic analysis. A description of the specific coding methodology and sample quotations would be helpful for the reader to understand the qualitative findings and how they were derived. 3. Inclusion of a Consort diagram would clarify how the final sample size was reached. 4. Descriptive characteristics that are presented in Table 1 do not need to be repeated in the description of the results, except for specific notable findings. Also, the authors may consider including p-values in Table 1.
--	--

VERSION 1 – AUTHOR RESPONSE

Reviewer: 1

Dr Simon Tso

Jephson Dermatology Center, South Warwickshire NHS Foundation Trust, United Kingdom Please state any competing interests or state 'None declared': I have previously published in the same field. Tso S, Potential barriers to pursuing a career in academic dermatology. 2016. 175(1):222-223. DOI: 10.1111/bjd.14519 Tso S. Clinical academic career: an alternative viewpoint. Clinical Teacher. (in press) doi: 10.1111/tct.12545

Please leave your comments for the authors below I read the manuscript with great interest which covers an important topic. However, prior to acceptance for publication, I would recommend the authors to consider my comments and recommendations on how this piece of work could be strengthened.

1) Statistics: please review all statistical data to ensure the % will add up to 100%, for example Page 9 Table 1 line 10-15 female respondents % added up to 101%. In other tables, the statistics added up to 99% or 101%.

In each of the tables where this occurs we acknowledge, in the footnotes, that not all percentages add to 100 due to rounding. The inclusion of decimal points would be necessary for the percentages all to add to 100. On discussion, we feel that this detail would not add additional information whilst impacting on the clarity of the tables. In this we follow the advice provided on the BMJ YouTube channel (<https://www.youtube.com/watch?v=B2FDbGUzNRI>).

2) Page 35 Box 5. Please provide justification for classifying specialties into craft and non-craft specialties. For example, the specialty I come from, Dermatology, many dermatology surgeons undertake Moh's micrographic surgery and undertake surgical fellowships. They would consider their work as a craft as well. Likewise, interventional cardiology colleagues may also feel the same way too. Otherwise, Box 5 (and all relevant results) should be presented in an alternate format.

Preliminary findings from focus groups conducted as part of Oxford's Clinical DPhil Paths study indicate that clinical doctoral students from specialties which require surgical skills may face challenges associated with maintaining such skills during the doctorate. We attempted to take this into account by splitting participants into craft and non-craft specialty groupings and using this variable in the analyses. However, we recognise that the split is imperfect since some medical specialties can involve conducting surgical procedures.

On reflection, we are unable to improve on how we performed the craft/non-craft split based on our data, and we do not feel that there is an alternative useful way of grouping the diverse specialties to

which participants belonged that respects some of the challenges that the reviewer identifies. Therefore, we have now removed references to the analyses involving comparisons between different types of specialty. Instead, we added to the Supplement an overview of respondents' specialty types (Box 4).

3) Qualitative analysis: There is insufficient information about the qualitative analysis method employed in this study to permit interpretation of the data. Looking at the results, I believe you were using content analysis to count the frequency of responses with similar codes/themes. The method used should be referenced where appropriate. How did the two universities compare the codes (face-to-face vs teleconference)?

We have now added additional information about the qualitative data analysis to Methods (p.7), and an in-depth description of the process to the Supplement (Box 3).

4) Study population: Were MD (research) students excluded from this study? If so, why? Did the study include suspended students? Were part time students included?

We have now edited the manuscript on pages 2, 6, and 18 to clarify our population and why MD/DM students were excluded. In brief, MD/DM students frequently do not have a structured programme of study compared to PhD students, and they can also be doing their research in a different institution to the one where they are registered. Taking these together, we felt that the heterogeneity of experience could mean that a number of the survey questions would be viewed differently to PhD respondents and may confuse the results.

We also now annotated the STROBE flow diagrams (added to the Supplement – see below) to further clarify that all enrolled clinical PhD students were surveyed, independently of their part- or full-time status.

5) Clinical academic career: Could the authors comment in the methods section on why the study narrowly chosen the role of 'Clinical Lectureship' as an end point when there are so many different clinical academic roles out there? You may wish to comment on the lack of clear career path in clinical academia

We concentrated on clinical lectureships as these are the first postdoctoral post in the integrated clinical academic training (IAT) pathway and increasingly important in the pursuit of a clinical academic career, as described in the Introduction leading up to our research aims. To assist readers with understanding why we concentrated on clinical lectureships, we have elaborated on this in the last paragraph of in the Introduction (under research aims).

We recognise that there are different clinical academic roles but we felt it important to focus on the formally recognised UK career pathway that has emerged in the last 10 years to explore what factors may affect long-term career plans in this regard.

6) Page 4 line 16 please change the word 'fears' to an alternative, for example, 'concerns'

We have now made this change.

7) Methods: page 5: how much time were given between the two / three email reminders? When was the online or paper questionnaire sent out?

We have now included this information (in the STROBE flow diagrams).

8) Page 4 line 40 Ref 9-11. You may wish to read the work by Bucklin et al 2014, Lyons et al 2010 and Alenxander and Lang (2008) which may help you to expand on the range of barriers and enablers contributing to the issue under investigation.

Bucklin BA, Valley M, Welch C, Tran ZV, Lowenstein SR. Predictors of early faculty attrition at one Academic Medical Center. BMC Med Educ 2014;14:27.

Lyons OTA, Smith C, Winston JS, Geranmayeh F, Behjati S, Kingston O, Pollara G. Impact of UK academic foundation programmes on aspirations to pursue a career in academia. Med Educ 2010;44(10):996–1005.

Alexander H, Lang J. The long-term retention and attrition of U.S. Medical School Faculty. Association of American Medical Colleges 2008;8:4.

Thank you for highlighting these. The papers suggested are certainly relevant to the research area, and indeed Bucklin et al.'s was reviewed by Ranieri et al. (2016), which we reference. We have now added a reference to Lyons et al. in the Introduction. However, having also read the other papers, we do not feel that there is a need to add these references to our manuscript; they concern attrition rates amongst US faculty and, whilst important, are not easily generalizable.

Overall, we feel that the references which we provide in the Introduction cover a very broad range of issues, particularly the ones that are useful for the Discussion of our findings later on in the paper.

Reviewer: 2

Danielle Rodin

University of Toronto, Canada

Please state any competing interests or state 'None declared': None declared.

Please leave your comments for the authors below This paper presents findings from a cross-sectional study of the factors that contribute to the career intentions of 320 medically qualified doctoral students at two research-intensive UK universities. This study was undertaken because of concerns about the retention of such trainees in academic medicine. It is the first study of its kind in the UK context and the first cross-sectional study on this topic. Based on a large sample size, the investigators found that the perception of limited opportunities and unsatisfactory work-life balance contributed to career intentions and that only half of the surveyed intended to pursue an academic career. Lack of information and exposure to opportunities affected outcomes, although it is notable that gender did not affect the outcomes assessed. A number of recommendations are made to increase the likelihood of an academic career intention.

Issues that the authors might wish to address include:

1. The authors include the questionnaire in the Appendix. However, it would be helpful if information was provided about how the questionnaire was developed, whether items were drawn from existing

questionnaires or from the literature, or whether the questionnaire has been validated in any way.

We have now added this information to the Supplement (Box 2, which contains the questions).

2. The study is predominantly quantitative, but the questionnaire includes several open-ended questions, from which responses were analyzed by thematic analysis. A description of the specific coding methodology and sample quotations would be helpful for the reader to understand the qualitative findings and how they were derived.

We have now added additional information about the qualitative data analysis to Methods (p.7), and an in-depth description of the process, including quotations, to the Supplement (Box 3).

3. Inclusion of a Consort diagram would clarify how the final sample size was reached.

We have now added diagrams, one for each of our census surveys, to the paper's Supplement (Box 1). Since we have reported our findings according to STROBE guidelines, we refer to them as "STROBE", rather than "CONSORT", diagrams.

4. Descriptive characteristics that are presented in Table 1 do not need to be repeated in the description of the results, except for specific notable findings. Also, the authors may consider including p-values in Table 1.

We have now reduced some text about participants' clinical training level on p.8 (but kept mention of the proportions who have completed training, since this is a key finding to which we return in the Discussion).

Thank you for suggesting significance testing for Table 1. We have considered it but, as the table is provided for descriptive purposes, in accordance with our aims of "describing the demographic and clinical-training characteristics of (...) clinical doctoral students [and] determine their long-term career plans" (p.4), we do not feel that the inclusion of p-values here would be helpful. Later in the paper (Table 2 and Box 6), we present the outcomes of our statistical tests with long-term career plans as the outcome variable, in accordance with our aim of determining which factors influence participants' long-term career plans (p.4).

VERSION 2 – REVIEW

REVIEWER	Dr Simon Tso Department of Dermatology, University Hospitals Coventry and Warwickshire NHS Trust, United Kingdom I have previously published in the same field. Tso S, Potential barriers to pursuing a career in academic dermatology. 2016. 175(1):222-223. DOI: 10.1111/bjd.14519 Tso S. Clinical academic career: an alternative viewpoint. Clinical Teacher. (in press) doi: 10.1111/tct.12545
REVIEW RETURNED	09-Jun-2017

GENERAL COMMENTS	Thank you for submitting your revised manuscript and addressing the issues I have raised from my previous review. It reads well and
---

	data is well presented. I have two further minor comments (see below). Well done with the revision. Comment 1: Page 89, Section heading: Implications of findings, line 24, clinical 'doctoral' students, should change to clinical 'PhD' students. Likewise page 90, section heading Conclusions and further research, line 25, 'doctoral students' should be changed to 'PhD students' and page 91, section heading acknowledgments, line 6, change clinical doctoral students to clinical PhD students. Please ensure the same terms/phrases are used throughout the manuscript Comment 2: Page 78, Section heading Results: Response rates: 'A more resource intensive approach may have yielded higher response rates. Consider putting this under discussion section rather than results section.
--	--

VERSION 2 – AUTHOR RESPONSE

Comment 1

Thank you for your comments. In the sections highlighted, we have replaced 'doctoral' students with 'PhD' students where 'doctoral' refers specifically to the students selected to take part in the study. In addition, we have also made this replacement in the Introduction (last paragraph), the Method (Instruments), the Results (first page) and in the Discussion (first page).

Comment 2

Thank you for raising this. We included this statement in response to an editorial request. In the interests of brevity we felt it helpful to mention this under the 'Response rates' heading rather than in the Discussion where we do not discuss the response rate. On reflection we feel that it is better to leave it where it is rather than introduce a new topic in the Discussion.